# Effective, Rapid, and Small-Scale Bioconjugation and Purification of “Clicked” Small-Molecule DNA Oligonucleotide for Nucleic Acid Nanoparticle Functionalization

**DOI:** 10.3390/ijms24054797

**Published:** 2023-03-02

**Authors:** Erwin Doe, Hannah L. Hayth, Ross Brumett, Emil F. Khisamutdinov

**Affiliations:** Department of Chemistry, Ball State University, Muncie, IN 47306, USA

**Keywords:** bioconjugation, click chemistry, ODN conjugate purification, fluorescence reporters, nucleic acid nanoparticles

## Abstract

Nucleic acid-based therapeutics involves the conjugation of small molecule drugs to nucleic acid oligomers to surmount the challenge of solubility, and the inefficient delivery of these drug molecules into cells. “Click” chemistry has become popular conjugation approach due to its simplicity and high conjugation efficiency. However, the major drawback of the conjugation of oligonucleotides is the purification of the products, as traditionally used chromatography techniques are usually time-consuming and laborious, requiring copious quantities of materials. Herein, we introduce a simple and rapid purification methodology to separate the excess of unconjugated small molecules and toxic catalysts using a molecular weight cut-off (MWCO) centrifugation approach. As proof of concept, we deployed “click” chemistry to conjugate a Cy3-alkyne moiety to an azide-functionalized oligodeo-xynucleotide (ODN), as well as a coumarin azide to an alkyne-functionalized ODN. The calculated yields of the conjugated products were found to be 90.3 ± 0.4% and 86.0 ± 1.3% for the ODN-Cy3 and ODN-coumarin, respectively. Analysis of purified products by fluorescence spectroscopy and gel shift assays demonstrated a drastic amplitude of fluorescent intensity by multiple folds of the reporter molecules within DNA nanoparticles. This work is intended to demonstrate a small-scale, cost-effective, and robust approach to purifying ODN conjugates for nucleic acid nanotechnology applications.

## 1. Introduction

The major practical purpose of nucleic acid nanotechnology in medicine is the application of nanoparticles as a drug delivery system, which is a fundamental part of drug development, and a wide range of drug delivery nano-vehicles has, thus, been designed [1,2]. Most of the new potential therapeutic molecules are currently lacking good pharmacokinetics and biopharmaceutical profiles [3,4]. The therapeutic properties can be potentially improved by more efficient delivery to their biological targets using the advantages of the emerging field of nucleic acid nanotechnology [5,6,7]. Additionally, some drugs that have previously failed clinical trials might also be re-examined using nanotechnological approaches [1]. Nucleic-acid-based multifunctional nano-delivery systems are advantageous as they utilize a modular approach to combine targeting, diagnostic, and therapeutic modules, as has already been exemplified by an astonishing number of existing reports [8,9].

Nucleic acid nanotechnology has often taken advantage of the unique properties of RNA molecules possessing gene-regulating functions (and, thus, serving as a drug component). Nucleic acid nanoparticles are designed to exhibit a new purpose, for example, to control therapeutic release, drug targeting, and fluorescence reporters [10]. However, to achieve these properties, the conjugation of ligands to nucleic acid oligonucleotides is often required. Due to progress in the chemistry of bioconjugation, various methods have been employed to link ligands with reactive groups on the surface of the nanocarriers [11,12,13,14]. These methods can be broadly classified into covalent and noncovalent conjugations. Noncovalent bonding, which proceeds by the relatively weak association of targeted ligands to the nucleic acid nanocarrier, is exemplified in the nucleic acid aptamers field [15,16,17]. Such an interaction has the advantage due to the avoidance of rigorous, destructive reaction agents. Covalent conjugation often involves interactions between two thiol groups—alkyne and azide, carboxylic acid and primary amine, maleimide and thiol, hydrazide and aldehyde, and primary amine and aldehyde. Among the above examples, the interaction between alkyne and azide groups has attracted particular attention, evidenced by a 2022 Nobel Prize award in chemistry “for the development of click chemistry and bioorthogonal chemistry” [18].

Indeed, click chemistry comes in handy as an indispensable synthetic tool for the efficient fabrication of nucleic-acid-based conjugates in the evolving field of oligotherapeutic research. Click chemistry was designed to generate reaction types that give very good yields, possess a wide range of applications, and give products that are stereospecific without necessarily displaying enantioselectivity [19,20,21]. The routes of the reaction must require simple reaction conditions and yield byproducts that do not require a chromatographic approach to purify [21,22,23,24]. Thus, click reactions have become the most popular of all bioconjugation techniques and have been exemplified in numerous studies as an efficient and easy-to-use bioconjugation method for the covalent coupling of molecules. Its wide scope has tremendously impacted many disciplines of scientific research, including biochemistry, where it has been largely used as an important coupling method in the nucleic acid field. Azide–alkyne cycloaddition reactions are among the most utilized click reactions by virtue of their success in modifying and functionalizing biological molecules with azide and alkyne groups.

By applying azide–alkyne cycloadditions, oligonucleotides have been successfully ligated, cyclized, and labeled efficiently and with little cost compared to other chemical conjugation techniques. For example, with the aid of copper-(I)-catalyzed azide–alkyne cycloaddition reactions, Chittepu et al. were able to successfully couple a fluorescence reporter (an azide-functionalized coumarin) to four nucleobases [25]. This successful tagging of nucleobases and, for that matter, nucleic acids makes it possible to track and study biochemical reactions in vivo. A terminal alkyne residue of DNA oligonucleotides was coupled to azide-functionalized nonfluorescent 3-azido-7-hydroxycoumarin to successfully generate a fluorescent product. Guo’s laboratory applied RNA nanotechnology to enhance the solubility of drugs that poorly dissolve in aqueous solutions, using camptothecin (CPT) to serve as proof of concept in the targeted delivery of anticancer agents [7,26]. Their research adopted copper-(I)-catalyzed azide–alkyne cycloaddition (CuAAC) to conjugate multiple CPT prodrugs to RNA oligonucleotides to form CPT-RNA conjugates. Their study showed that the CPT-RNA nanoparticles exhibited very good solubility in an aqueous medium as well as very effective cell binding and internalization, resulting in the death of tumor cells [7].

This paper aims to introduce a general, benchtop-compatible, and rapid approach to “click” and purify commercially available ODNs with small molecules to aid the advancement of nucleic acid nanotechnology in therapeutic fields. In particular, we examined the applicability of MWCO membrane to achieve high purity of the conjugated products and further decorate triangular-shaped DNA nanovehicles with fluorescently labeled conjugates. The fabrication of pharmaceuticals that involves copper-based click reactions poses a potential threat to mediating the generation of reactive oxygen species in living cells if the copper is not removed [27,28]. The cytotoxic threat of catalyst-based click reactions to living systems was a significant challenge that inspired Carolyn Bertozzi and colleagues to engineer the use of cyclooctyne to execute copper-free click reactions in living systems upon the modification of Staudinger ligation [29]. Implementing the MWCO device, we intend to demonstrate a simple in vitro approach to eliminating the unwanted cytotoxic copper after a click reaction. The oligomers are clicked with Cy3-alkyne and 3-azido-7-hydroxycoumarin to form DNA-Cy3 and DNA-Cou products. A molecular weight cut-off membrane filter 3 KDa is used as a rapid means of purifying the clicked oligomers (conjugates), after which the fluorescence spectra of crude conjugates (unpurified), retentates (purified conjugates), and filtrates are collected and compared to investigate yields. Both sets of collected spectra demonstrate a prominent emission intensity for the purified conjugates and purified hybridized nanoparticles relative to filtrates and all controls used.

## 2. Results

### 2.1. Conjugation of ODN Alkyne with 3-Azido-7-hydroxycoumarin

Profluorogenic dye 3-azido-7-hydroxycoumarin (referred to as coumarin azide in the text below) exhibits no intrinsic fluorescent properties until the molecule reacts with a terminal alkyne, resulting in the formation of an intense fluorescent 1,2,3-triazole product [30]. This property has proved to be a versatile tool in the application of labeling of various biomolecules and the combinatorial synthesis of fluorescence dyes due to its high reaction efficiency in mild conditions [29,31]. Taking advantage of this fluorogenic molecule, we first performed the optimization of the “click” reaction with coumarin, as described in the Materials and Methods, with a slight deviation from the previously reported protocols [31,32] Figure 1A. The successfully conjugated product can be readily observed with the naked eye by exposing the reaction vessel to a long wavelength UV lamp (365 nm). Figure 1B demonstrates a bright, cyan-colored light emitting from the post-conjugation reaction test tube. We found that two times the excess amount of coumarin azide to ODN is optimal to achieve efficient conjugation. Further analysis of the products by 32% denaturing PAGE showed that the conjugate (DNA-Cou) migrates slower than the unconjugated ODN due to the molecular weight differences (Figure 1C). The gel image, obtained prior to staining with ethidium bromide (EB), shows a detectable emission signal of the reaction product in contrast to control DNA alkyne and Cou-azide lanes. The DNA alkyne band was visualized after staining the gel in EB solution. No bands were observed for coumarin azide in both cases due to its uncharged state; the molecule was not affected by electrophoretic force and, hence, did not enter the gel matrix.

To calculate the conjugation yield, we used gel band quantification by Image J software [33,34]. Identical rectangular boxes were drawn around the DNA alkyne and DNA-Cou bands using the above quantification software. The resulting integrated band intensity of the DNA-Cou band was compared to the DNA alkyne band treated as 100%. This method is generally accurate, considering the amount of the DNA only and DNA-conjugate was identical per individual well. A yield of 86.0 ± 1.3% was obtained after several rounds of quantification analysis on three repetitive gel experiments.

### 2.2. Conjugation of ODN-Azide with Cy3-alkyne

The identical “click” reaction was implemented to conjugate ODN-azide to Cy3-alkyne (Figure 2A). The resulting 32% denaturing gel analysis of the “clicked” product, DNA-Cy3, clearly possesses retardation of band migration compared to the control lanes DNA azide and Cy3-alkyne, indicating the success of the reaction (Figure 2B). When the gel was imaged prior to staining with EB, no band was observed for DNA azide, and it became visible after staining. Cy3-alkyne, being an organic fluorescent dye, was observed after imaging prior to staining with EB. The yield of the conjugated product was found to be 90.3 ± 0.4% using the quantification approach by Image J.

Collectively, the “click” protocol for the 12-mer ODN and small fluorescence molecules can be potentially utilized to conjugate various other molecules, achieving high efficiency. However, to further utilize the ODN conjugates in nanoparticle assembly for in vitro and in vivo applications, it is critical to purify the products to remove any excess of the unconjugated small molecules as well as the toxic Cu(I) catalyst [35,36]. Generally, chromatography methods, such as size exclusion and ion exchange, work well in this regard; however, alternative, cost-effective approaches that save materials and time are highly demanded in the chemical conjugation discipline in general [37]. Below, we describe an alternative approach to rapidly purify the conjugated products using the advantages of a molecular weight cut-off filtering device.

### 2.3. Purification of the DNA-Conjugates Using MWCO Spin Column

Rigorous attention to purification methodologies during the synthesis and fabrication of nucleic acid nanoparticles is essential to achieving reproducible and well-defined performances in the application of nucleic acid nanoparticles as therapeutic agents. The purification of biological molecules based on desalting and buffer exchange using an MWCO membrane has been well established and widely used in the protein field. Early membranes were used primarily to concentrate and purify protein solutions to retain certain-sized proteins. The ability of the membrane to retain biomolecules is based on their molecular masses. The term MWCO is, therefore, often defined as the molecular weight at which 90% of the solute is passed through (rejected by) the filter membrane when applying centrifugal force.

The calculated molar masses of the DNA-coumarin and DNA-Cy3 are 3960.65 g/mol and 4722.8 g/mol, respectively. When applying their solutions to the 3 KDa MWCO ultrafiltration system (Millipore Sigma, St. Louis, MO, USA), it is expected that all unconjugated reactants, including coumarin azide (MW = 203.15 g/mol) and Cy3-alkyne (MW = 964.31 g/mol), as well as the Cu(I) catalyst and other low molecular weight substances will be rejected by the membrane, as illustrated in the schematic diagram in Figure 3A. The fluorescence properties of the conjugated molecules were determined by measuring their fluorescence emissions before and after ultracentrifugation to track the progress of purification. The spectra shown in Figure 3B,C demonstrate fluorescence signals of the DNA-coumarin and DNA-Cy3 reaction products taken before (black line) and after three rounds of filtrations (red-colored line). After each round of filtration, the retained volume of reaction buffer was supplied with TEAA buffer, pH = 7.0, thus, facilitating buffer exchange. The purified ODN conjugates exhibited relatively higher intensity maxima compared to the crude products. In contrast to coumarin azide, the Cy3-alkyne exhibits a fluorescence signal regardless of the formation of the 1,2,3-triazole product. Both the crude and purified DNA-Cy3 elicited a more prominent emission intensity relative to the Cy3-alkyne and the filtrate (Figure 3C). The emission observed for the filtrate was, most likely, due to the excess of unconjugated Cy3-alkyne used in the click reaction. The concentration used for the positive control was the same as the concentration of the Cy3-alkyne used in the click reaction, and therefore, it showed a higher emission compared to the filtrate (excess Cy3-alkyne obtained in the filtrate after purification with the filtering device). The excess Cy3-alkyne could not have exhibited a higher emission intensity than the positive control because part of the excess Cy3 was involved in the click reaction for the formation of the clicked product (DNA-Cy3). The significant difference in emission intensity of the clicked product versus the positive control and the filtrate is presumably because of the formation of the triazole moiety, which serves as a fluorophore to the Cy3 core structure in the clicked product.

Based on the fluorescence analysis of the filtration fractions, it becomes evident that this is a very efficient approach and can be used not only for purification purposes but also for buffer exchange. Notably, the residual amount of unconjugated DNA with a molecular weight of around 3800 Da is expected to be found in the retentate fraction. However, considering the high conjugation efficiency of the “click” reaction, at around 90%, the presence of an insignificant amount of unconjugated DNA should not affect the assembly efficiency of the nucleic acid cargo. Furthermore, because the azide and alkyne groups are considered to be biorthogonal, it is highly unlikely they will interact with biological entities [38] once delivered into the cell.

### 2.4. Decorating Triangular DNA Nanoparticles with ODN Conjugates

To further test the applicability of the ODN coupled with fluorescence dyes, we assembled a DNA triangular nanoparticle that was previously developed in our lab based on the tetra-T motif [39]. Two sets of triangular DNA nanoparticles containing sticky ends were self-assembled in the presence of increasing concentrations of ODN-Cou or ODN-Cy3 1:0, 1:1, 1:2, and 1:3 molar ratios, as exemplified in the secondary structure in Figure 4A. The length of the sticky end nucleotides was designed to be long enough to sustain stability under physiological temperature. The measured melting temperature (Tm) of the 12-base pair duplex was found to be 57 ± 0.3 °C, indicating modest thermal stability (Figure 4B). The resulting DNA nanoparticles were subjected to 6% native gel electrophoresis to evaluate their relative migrations. Upon increasing the molecular weight of the DNA constructs from 1:0 to 1:3 (the triangle to ODN-dye ratio), a clear band mobility retardation was observed for both triangles harboring ODN conjugates, indicating a strong interaction between ODN-dye and triangle nanoparticles (Figure 4C). The gel shift assay, however, does not provide sufficient information about the fluorescence properties of the resulting nanoparticles. Therefore, we further conducted fluorescence experiments of the assembled DNA complexes in solutions. Figure 4D shows the fluorescence properties of the Tri-DNA-Cou and Tri-DNA-Cy3 nanoparticles in various ratios (1:0, 1:1, 1:2, and 1:3) (Figure 4C). The 1:0 Tri-DNA-Cou served as the negative control. An increased emission intensity at λ_max_ = 477 nm was observed for the DNA decorated with various ODN-coumarin in the order 1:0, 1:1, 1:2, and 1:3, from lowest to highest. Similarly, for the triangle decorated with ODN-Cy3, the highest intensity observed for the 1:3 ratio was captured at λ_max_ = 570 nm Figure 4C. The combined results from native PAGE and fluorescence assays indicate that ODN-dye conjugates obtained by click chemistry and purified through MWCO ultracentrifugation can be readily applied for DNA nanoparticle assembly to fabricate functional nanovehicles for bioimaging purposes.

## 3. Discussion

Small-molecule therapeutic agents have been used in the treatment of many health conditions. However, their use encounters a plethora of challenges, including poor solubility, as well as inefficient delivery to target sites. Click chemistry is an emerging field beaming with prospects, especially for biological and biochemical sciences. Many coupling reactions have been executed using click chemistry as the key tool to fashion many molecules, nanomaterials, etc., which find applications in drug delivery, bioimaging, and other areas of nanotechnology. Click chemistry makes it easy to manipulate and modify biological molecules for coupling to desired molecules to solve complex problems with “easy” steps. Following “click”-based coupling reactions, purification techniques would be capable of separating the ODN-conjugate from both unconjugated small molecules and ODNs. Thus, demanding attention to purification and characterization methodologies during the design and assembly of nucleic-acid-based nanoparticles is essential to achieving reproducible and well-controlled performance in the intended application. Often, chromatography methods with their multiple variations are considered the “gold standard”, and this is the most common approach applied to purify bioconjugates. However, each chromatography method requires condition optimization for a particular analyte, the use of sophisticated instrumentation, columns, and certain chemicals in the mobile phase to either provide desired pH or hydrophobicity, making the chromatography method laborious and time-consuming.

The membrane filtration method (including dialysis) is relatively inexpensive, readily available from multiple commercial sources, and intuitive to use. The method uses a semipermeable membrane with a variation in pore sizes or molecular weight cut-off values to separate unconjugated small molecules from the DNA-conjugates. Ultrafiltration devices are generally driven by centrifugal forces to increase the speed and efficiency of the purification process, saving laboratory time and effort [40]. Surprisingly, to the best of our knowledge, there is no, or only limited, information available where researchers use ultracentrifugation devices to purify clicked oligonucleotides to expunge unconjugated reactants, including copper ions, which are potential cytotoxins to living systems. To date, ultrafiltration devices have shown some utility for example to concentrate nucleic acid nanoparticles extracted from gels to prepare samples for atomic force microscopy [41,42].

In this work, we developed a small-scale ODN fluorescence dyes coupling method using a Cu^I^-catalyzed “click” reaction with a rapid purification methodology that is generally applicable to a wide range of other small molecules. We successfully demonstrated ODN coupling efficiencies of ~90%, with both Cy3 and coumarin molecules possessing strong fluorescence emission after filtration and after hybridization with the DNA nanoparticles. The clicked products can be purified and desalted by using a filtering device with a 3 KDa molecular weight cut-off membrane. The DNA-conjugates were successfully implemented to assemble fluorescence DNA nanoparticles with controlled stoichiometry. As this work is currently progressing, we envision the implementation of the assembled DNA triangular nanoparticles containing fluorescence dyes, with fine-tuned stoichiometry, for quantitative imaging applications in vitro and in vivo.

## 4. Materials and Methods

### 4.1. DNA Stock Solutions and Buffer Compositions

Lyophilized azide and alkyne-functionalized 12-mer ODNs were purchased from IDT technologies at 5 μmol synthetic scale:/5AzideN/CGCGCTCTTACG-3′and/5Hexynyl/CGCGCTCTTACG-3′. ODNs were dissolved in double-deionized water (ddH_2_O, Millipore) to prepare 2 mM solutions.

DNA strands for triangular nanoparticle assembly were obtained from IDT at 1 μmol synthetic scale and resuspended in ddH_2_O to prepare stock 100 μM solutions (Table 1).

Tris–borate–EDTA (TBE) buffer (pH = 8.0, 89 mM Tris base, 86 mM boric acid, and 2 mM EDTA); Tris–Acetate–EDTA (TAE) buffer (pH = 8.0, 40 mM Tris base, 20 mM acetic acid, and 1 mM EDTA); Tris–borate–magnesium (TBM) buffer (pH = 8.0, 89 mM Tris base, 86 mM boric acid, and 5 mM MgCl_2_.); and Tris–magnesium–saline (TMS) buffer (pH = 8.0, 50 mM Tri-HCl, 100 mM NaCl, and 10 mM MgCl_2_.) were prepared from the corresponding analytical grade chemicals (Millipore Sigma).

### 4.2. Stock Solutions for Click Chemistry

Ascorbic acid 5 mM was prepared in 20 mL of ddH_2_O to serve as the reducing agent for copper (II) to copper (I) in situ in the click reaction of the oligomers. A 2 M stock of triethylammonium acetate (TEAA) click reaction buffer (pH, 7.0) was prepared by mixing 2.78 mL of triethylamine and 1.14 mL of glacial acetic acid (AA). This mixture was topped up with ddH_2_O to make 10 mL of solution, and the pH of the resulting solution was adjusted to 7.0 using either triethylamine or AA. A copper-(II)-stabilizing complex, which was prepared by dissolving 0.025 g of CuSO_4_.5H_2_O in 10 mL ddH_2_O and mixed with a solution of TBTA (Tris [(1-benzyl-1H-1,2,3-triazol-4-yl)methyl]amine) made by dissolving 58 mg of the TBTA in 11 mL of ~100% dimethyl sulfoxide (DMSO). The resulting solution was vortexed to obtain a homogenous mixture. A 10 mM stock solution of 3-azido-7-hydroxycoumarin (coumarin azide) (203.15 g/mol, Jena Bioscience, Jena, Germany) was prepared in DMSO. The 10 mM Cy3-alkyne (MW = 964.31 g/mol, Millipore Sigma) stock solution was made using ddH_2_O.

### 4.3. General Procedure of Conjugation of DNA Alkyne to Coumarin Azide and DNA Azide with Cy3-alkyne

A total of 50 µL of 2 mM DNA alkyne was pipetted into a clean 1.7 mL Eppendorf tube supplied with a magnetic stir bar followed by the addition of 330 µL of 2 M TEAA buffer (pH, 7.0) and 450 µL of DMSO. The resulting mixture was vortexed to mix thoroughly. An amount of 20 µL of 10 mM coumarin azide stock solution was added to the mixture followed by the addition of 100 µL of a freshly prepared 5 mM ascorbic acid solution. The resulting mixture was vortexed followed by bubbling argon gas through the solution for 2–3 min to evacuate dissolved O_2_. Fifty microliters of copper (II)–TBTA complex solution was pipetted into the reaction mixture, and the resulting solution was degassed for an additional 1 min. The reaction mixture was kept overnight over a magnetic plate at room temperature with a moderate spinning rate in the dark.

The same procedure and proportions were followed for the conjugation of DNA azide with Cy3-alkyne. Cy3-alkyne was used as a substitute for coumarin azide.

### 4.4. Denaturing Polyacrylamide Gel Electrophoresis (PAGE)

Four microliters of the clicked product (DNA-Cy3 or DNA-Cou) was pipetted into a clean, appropriately labeled Eppendorf tube supplied with 6 µL of ddH_2_O. Ten microliters of 6 M urea loading dye was added to the samples and heated to 99 °C for 1 min, then snap-cooled on ice. Control samples (unconjugated DNA, Cy3-alkyne, or coumarin azide) were prepared in a similar manner as the clicked sample. A volume of 3 µL of each sample was placed into their respective wells of polyacrylamide gel (32% acrylamide, 37.5:1 acrylamide: bisacrylamide, 8 M Urea). The electrophoresis was conducted at a constant 130 V for 150 min at ambient temperature. The gel was imaged before and after staining with ethidium bromide.

### 4.5. Purification of Clicked Products Using Molecular Weight Cut-Off (MWCO) Membrane

Amicon ultra 0.5 mL centrifugal filters with 3 KDa molecular weight cut-off membrane based on regenerated cellulose (Millipore Sigma) were used as a filtering device to eliminate compounds with a molecular weight below 3 KDa and to retain the clicked products in the membrane after centrifugation. The crude product (400 µL) was pipetted into the filter membrane and centrifugation was executed at 14,000× *g* for 10 min. About 200 µL of 2 M TEAA buffer was added to the retentate, and the centrifugation was repeated for another 10 min. The last step was repeated one more time, and both filtrate and retentate were kept for fluorescence assessment.

### 4.6. Fluorescence Assay of ODN Conjugates before and after Purification

Emission spectra were collected on an FP-8350 Spectrofluorometer (JASCO) with the following settings: DNA-Cy3 conjugate, excitation wavelength set at 553 nm, and emission collected in the range from 560 to 700 nm; DNA-coumarin conjugate, excitation wavelength set at 404 nm, and emission scanned in the range from 414 to 650 nm. In all experiments, the integration time was 1 s with a detector voltage of 400 V. Signal intensities were registered as a count per second (cps). Excitation and emission slits were both set to 2 nm.

### 4.7. DNA Nanoparticle Self-Assembly with the DNA-Coumarin and DNA-Cy3 Conjugates

Self-assembly of DNA sequences of triangular nanoparticles was designed following our previous report [39]. Equimolar concentrations of individual strands (1 µM) were mixed in presence of various concentrations of conjugated ODNs in TMS buffer. To facilitate hybridization, the solution mixture of DNAs was heated to 80 °C for 5 min and slowly cooled down to 4 °C at a rate of 1 °C/min. An equal volume of loading buffer (sucrose 40% (*wt*/*vol*), 0.1% (*wt*/*vol*) xylene cyanol, and 0.1% (*wt*/*vol*) bromophenol blue) was added to each sample with subsequent analysis on 6% native PAGE in TBM. The gels were run at 60 V for 60 min at room temperature and imaged using ChemiDoc XRS (BioRad, Hercules, CA, USA) or Amersham Typhoon-5 (Cytiva, Marlborough, MA, USA) imaging systems with and without the presence of ethidium bromide.

### 4.8. DNA Duplex UV-Melting Assay

Complementary DNA strands 5′-CGCGCTCTTACG-3′/5′-CGTAAGAGCGCG-3′ were mixed in 100 µL TMS buffer to achieve 1 µM concentration. Absorbance versus temperature profiles were measured using a 1 cm path length cuvette (8 series Micro Cell, Shimadzu, Kyoto, Japan) on a Shimadzu UV-Vis spectrophotometer (UV-2600i model) equipped with Thermal Melt System TMSPC. Absorbance wavelength was recorded at 260 nm with a temperature ramp of 1 °C/min. Four consequent runs were performed covering the range of 20–90 °C to obtain average melting point. The graphical software OriginPro (OriginLab TM) was used to fit the obtained data by using a dose–response sigmoidal curve fitting model, as determined by Johnson, M.B., et al. [43].

## Figures and Tables

**Figure 1 ijms-24-04797-f001:**
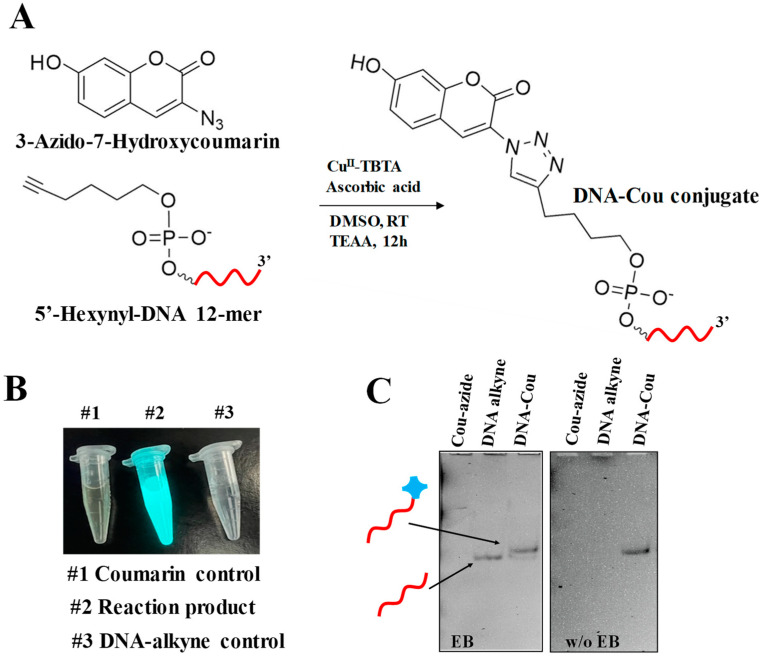
Coupling of ODN alkyne with coumarin azide utilizing “click” chemistry and product analysis. (**A**) Generalized reaction diagram interaction between 3-azido-7-hydroxycoumarin and 5′-Hexynyl-DNA in a 2:1 ratio in the presence of Cu(I) catalyst. (**B**) Irradiation of the DNA-Cou conjugate product and control samples by UV lamp 365 nm. (**C**) Denaturing 32% PAGE demonstrating band shifts between DNA alkyne (reactant) and DNA-Cou (product).

**Figure 2 ijms-24-04797-f002:**
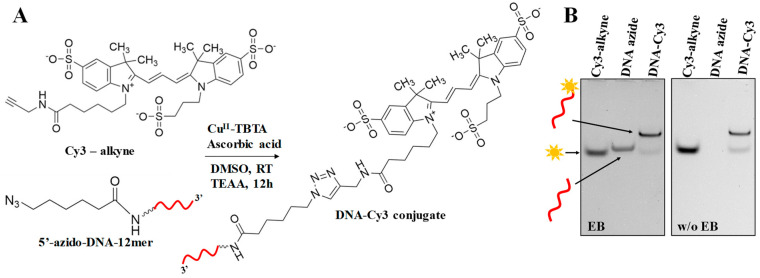
Coupling of 5′-azido-DNA (ODN-azide) with Cy3-alkyne by “click” chemistry and product formation investigation. (**A**) Generalized reaction diagram interaction between Cy3-alkyne and 5′-azido-DNA in 2:1 ratio in the presence of Cu(I) catalyst. (**B**) Denaturing 32% PAGE demonstrating band shifts between reactants (Cy3 and DNA azide) and product (DNA-Cy3).

**Figure 3 ijms-24-04797-f003:**
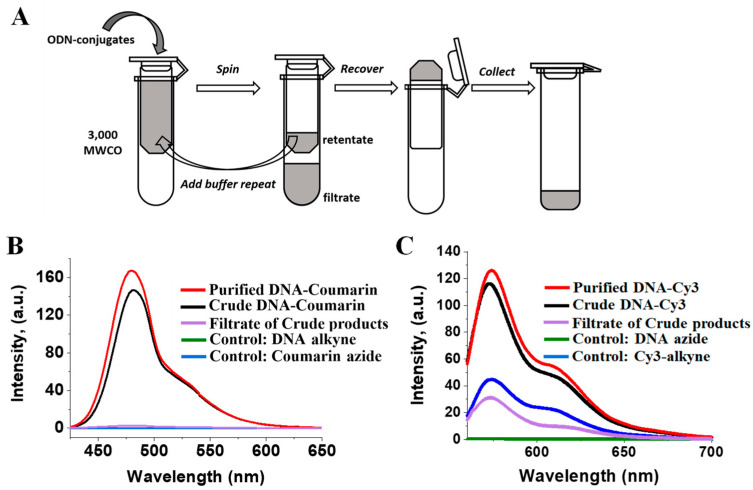
Illustration of membrane filtration principle based on MWCO centrifugal device and fraction analysis. (**A**) Schematic diagram of the centrifugation steps to evacuate unconjugated reactants and buffer exchange of the post-reaction mixture. Fluorescence assays of the collected fraction of the DNA-coumarin (**B**) and DNA-Cy3 (**C**) coupling reactions.

**Figure 4 ijms-24-04797-f004:**
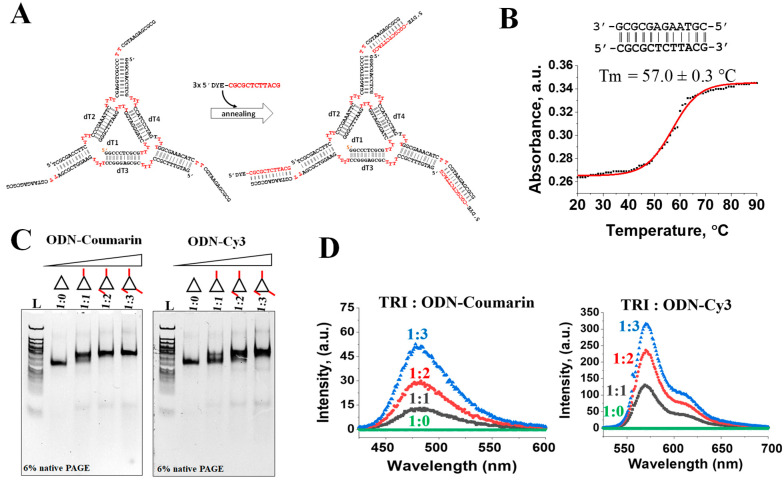
Hybridization of ODN conjugates with triangular-shaped nanoparticles. (**A**) Secondary structure and sequences of the DNA nanoconstruct representing three sticky-end regions complementary to the sequence of DNA-dyes. (**B**) Temperature-induced UV-absorbance profile of 1 µM DNA 12-base pairs in TMS buffer. Error corresponds to standard deviation in the mean. (**C**) Native PAGE analysis of the hybridization products upon saturation of the triangle nanoparticles with increasing number of DNA-conjugates to satisfy 1:0, 1:1, 1:2, and 1:3 stoichiometry ratios. (**D**). Fluorescence spectra of the triangular nanoparticles containing different numbers of the DNA-coupled dyes.

**Table 1 ijms-24-04797-t001:** DNA sequences for assembly of triangular nanoparticle. The underlined nucleotides correspond to sticky end regions complementary to the “Clicked” ODN.

Name	Sequence 5′ −> 3′
dT1	GGCCCTCGCG TTTT CTAGGGATGG TTTT GAATTTCGGG
dT2	TCGCGACCTTC TTTT CCCGAAATTC TTTT CGAGGTCGCCC TT **CGTAAGAGCGCG**
dT3	GATGTTTCGCC TTTT CGCGAGGGCC TTTT GAAGGTCGCGA TT **CGTAAGAGCGCG**
dT4	GGGCGACCTCG TTTT CCATCCCTAG TTTT GGCGAAACATC TT **CGTAAGAGCGCG**

## Data Availability

Data is contained within the article.

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
