# Peer review of "Effective, Rapid, and Small-Scale Bioconjugation and Purification of “Clicked” Small-Molecule DNA Oligonucleotide for Nucleic Acid Nanoparticle Functionalization"

_ijms, 2023, doi:10.3390/ijms24054797_

Round 1

Reviewer 1 Report

The manuscript titled “Effective, Rapid, and Small-scale Bioconjugation and Purification of “Clicked” Small Molecule-DNA Oligonucleotide for Nucleic Acid Nanoparticles Functionalization” by Erwin Doe et al. introduced a new application perspective of a molecular weight cut-off membranes (MWCO) as a purification tool for short single-stranded DNA coupled to small organic dyes. The authors demonstrated the implication of using a 3KDa MWCO device to filter out unconjugated reactants in a very effective and single step. Although the MWCO membrane technology has been available for several decades, there is a lack of information on the application of this centrifugation-driven filtration in bio-conjugation chemistry, which makes this article especially attractive. Also, the topic of bio-conjugation using click chemistry and subsequent “work-up” protocol is vital in the therapeutic oligonucleotide field. With the recent advances in silencing and antisense RNA therapeutic, there is a growing demand for developing efficient, cost-effective, scalable methods to achieve high purity of oligonucleotide-based drugs. In my opinion, this research article will enhance the interest of the scientific community in the IJMS journal and fits well to its scope. The data support the conclusion, the article is clear, organized, and well-written, and overall, I recommend this work for publication with a minor revision:

It will be informative to evaluate the stability of the hybridization of the ODN-conjugate 12-mer to the DNA triangular nano-partilces. A simple UV-melting experiment could provide information about the thermodynamic stability of the annealed 12-mer duplex.

Reviewer 3 Report

The manuscript describes the synthesis and purification of small molecules-DNA oligonucleotide conjugates, and their applicability to the assembly of DNA nanoparticles. Aim of the work, background information, experimental details and discussion of the results were clearly described.

Here are some small corrections/questions:

1.     The MWCO approach has been utilized for DNA nanoparticles previously. It is not clear what improvement this work will bring to the field.  

2.     How did the authors check the purity and identity of the products?

3.     Word or phrase has to be defined initially, so that the use of abbreviations later in the text would not confuse the readers.

4.     Some references are missing page numbers.

Round 2

Reviewer 2 Report

Thank you for improving your manuscript and addressing revisions promptly.

Reviewer 3 Report

Thank you for making the suggested changes.